Pre- and post-experimental manipulation assessments confirm the increase in number of birds due to the addition of nest boxes

Cuatianquiz Lima Cecilia 1 2
Macías Garcia Constantino 2 maciasg@unam.mx
1 Centro Tlaxcala de Biología de la Conducta, Universidad Autónoma de Tlaxcala , Tlaxcala , Mexico
2 Instituto de Ecología, Universidad Nacional Autónoma de México , Ciudad de México , Mexico
Renaut Jenny
Electronic publication date: 2016 Mar 15
Publication date: 2016
Volume: 4
Electronic Location ID: e1806
Received 2015 Nov 7; Accepted 2016 Feb 23
Copyright: ©2016 Cuatianquiz Lima and Macías Garcia
Copyright year: 2016
Copyright holder: Cuatianquiz Lima and Macías Garcia
License: This is an open access article distributed under the terms of the Creative Commons Attribution License, which permits unrestricted use, distribution, reproduction and adaptation in any medium and for any purpose provided that it is properly attributed. For attribution, the original author(s), title, publication source (PeerJ) and either DOI or URL of the article must be cited.
License URL: https://creativecommons.org/licenses/by/4.0/

Keywords: La Malinche, Secondary cavity-nesters, Nest-site availability, Forest management, Trans-Mexican volcanic belt

Funding: CONACyT -the Mexican Science Council Cecilia Cuatianquiz Lima received a PhD scholarship from CONACyT -the Mexican Science Council- during the elaboration of this work. The funders had no role in study design, data collection and analysis, decision to publish, or preparation of the manuscript.

==============================
Secondary cavity nesting (SCN) birds breed in holes that they do not excavate themselves. This is possible where there are large trees whose size and age permit the digging of holes by primary excavators and only rarely happens in forest plantations, where we expected a deficit of both breeding holes and SCN species. We assessed whether the availability of tree cavities influenced the number of SCNs in two temperate forest types, and evaluated the change in number of SCNs after adding nest boxes. First, we counted all cavities within each of our 25-m radius sampling points in mature and young forest plots during 2009. We then added nest boxes at standardised locations during 2010 and 2011 and conducted fortnightly bird counts (January–October 2009–2011). In 2011 we added two extra plots of each forest type, where we also conducted bird counts. Prior to adding nest boxes, counts revealed more SCNs in mature than in young forest. Following the addition of nest boxes, the number of SCNs increased significantly in the points with nest boxes in both types of forest. Counts in 2011 confirmed the increase in number of birds due to the addition of nest boxes. Given the likely benefits associated with a richer bird community we propose that, as is routinely done in some countries, forest management programs preserve old tree stumps and add nest boxes to forest plantations in order to increase bird numbers and bird community diversity.

Introduction

Worldwide, 26% of land birds rely on the presence of tree cavities in the environment to nest or roost (Newton, 1994; Newton, 1998). Of these, some dig cavities de novo (primary cavity excavators) whereas others (secondary cavity nesters) depend on pre-existing cavities that either form naturally or are previously dug by primary cavity excavators, mainly woodpeckers (Swallow, Gutierrez & Howard, 1986; Brawn & Balda, 1988; Newton, 1994; Martin & Eadie, 1999; Martin, Aitken & Wiebe, 2004; Lohmus & Remm, 2005; Remm, Lohmus & Remm, 2006) or other birds (e.g., European bee-eaters; Casas-Crivillé & Valera, 2005). Additionally, the presence of other organisms contributes to the creation of cavities (fungi, insects, amphibians, reptiles and mammals), and some secondary cavity nesters can modify the cavity to suit their needs, especially in soft wood (Newton, 1994; Martin, Aitken & Wiebe, 2004; Martin, Norris & Drever, 2006; Sánchez, Cuervo & Moreno, 2007; Lambrechts et al., 2010). Nevertheless, cavities are often a limiting resource and their availability and suitability can drive population processes in the species that use them, particularly among secondary cavity nesters (Brawn & Balda, 1988; Newton, 1994; Martin, Aitken & Wiebe, 2004; Aitken & Martin, 2012). Since cavity nesting species often have specific requirements, cavities that are appropriate for one species may not be suitable for others, thus increasing the intensity of both intra and interspecific competition (Dhondt, 2012).

Some researchers report that the availability of suitable cavities depends on the attributes of the trees present in a given area, including their size, age, architecture, hardness and density (Van Balen et al., 1982; Martin, Aitken & Wiebe, 2004; Martin, Norris & Drever, 2006; Sánchez, Cuervo & Moreno, 2007; Cornelius et al., 2008; Lambrechts, Schatz & Borgault, 2008; Cockle, Martin & Drever, 2010; Lambrechts et al., 2010). Several studies have shown that in undisturbed forest the number of suitable nesting cavities increases with tree age, with snags being an important source of nesting holes (Newton, 1994), such that in some communities secondary cavity nesters rely mainly on cavities in decaying trees (Gibbons et al., 2002; Remm, Lohmus & Remm, 2006; Wesolowski, 2007).

Since the density of cavities is positively related to tree density and age (Van Balen et al., 1982), secondary cavity nesters may be less likely to find nesting sites in managed forests, from where snags and old trees are often removed (Martin & Li, 1992; Newton, 1994; Holt & Martin, 1997; Martin & Eadie, 1999; Martin, Aitken & Wiebe, 2004; Saab, Dudley & Thompson, 2004; Sánchez, Cuervo & Moreno, 2007; Cornelius, 2008; Castro, Moreno-Rueda & Hódar, 2010; Goodenough, Elliot & Hart, 2009). Suitable cavities are also scarcer in young than in old woods, which are more structurally complex and have more cavities (Brawn & Balda, 1988; Waters, Noon & Verner, 1990). To mitigate the effect of lack of nesting cavities on bird communities, nest boxes are often placed in forest plantations where they may also favour the establishment of other cavity-dependent species such as mammals, amphibians, reptiles (Newton, 1994) and as many as thirty-nine invertebrate taxa, notably paper wasps and spiders (Mccomb & Noble, 1982). Indeed, many studies report greater densities of cavity-nesting birds in places where nest boxes have been added (Brush, 1983; Brawn & Balda, 1988; Waters, Noon & Verner, 1990; Sánchez, Cuervo & Moreno, 2007; Miller, 2010; Dhondt, 2012). However, few reports include an initial count of either birds or cavities (Brawn & Balda, 1988; Miller, 2010; Aitken & Martin, 2012), thus the possibility that bird numbers differed between sites before the addition of nest boxes cannot always be ruled out. Consequently, properly controlled experimental manipulations are needed to determine the nature and extent of the effect of nest boxes on secondary cavity nesters (Brawn & Balda, 1988; Dhondt, 2012).

Here we report on an experimental manipulation in temperate Mexican forest where we (1) assessed nest cavity availability, (2) added next boxes to trees in sampling points, and (3) counted the abundance of secondary cavity nesters and of breeding pairs both before and after the addition of nest boxes. We conducted this study at the La Malinche National Park (LMNP), which encompasses the upper part of La Malinche, an inactive volcano in central Mexico. In this area a patchwork of forest management practices afforded a variety of environments in which to test our predictions that there would be fewer cavities and secondary cavity nesting birds in young than in a mature forest, and that adding nest-boxes to both forest types would increase the numbers of secondary cavity nesters, especially in young forest.

Methods

Study area and experimental design

With an extension of 45,711 ha, the La Malinche National Park is the most important protected area in the state of Tlaxcala (Fig. 1). It harbours great biological diversity, including 27 mammal, 69 bird, 11 reptile, and five amphibian resident species, and an abundance of Dikarya (formerly Deuteromycota) fungi, Amebozoa (Myxomicetes) and plants. At least six secondary cavity nesting bird species belonging to five families nest at La Malinche National Park; both Pygmy (Sitta pygmaea) and White-breasted Nuthatches (Sitta carolinensis), House Wrens (Troglodytes aedon), brown creepers (Certhia americana), Mexican Chickadees (Poecile sclateri) and Western Bluebirds (Sialia mexicana; Howell & Webb, 2005; Windfield, 2005). Fifty-one percent (=23,612 ha) of the official park area has been claimed by local communities for growing crops (mainly maize) and to expand their urban areas. The other half of the park is covered by young deciduous (mostly below 2,800 m.a.s.l.), young coniferous (restoration forest), or mixed mature forest (Villers & López, 2004). We established study plots in two contrasting vegetation types; mixed mature forest, and forest at an advanced restoration stage of uniformly young conifer trees. Each vegetation category was determined by both the species composition and the diameter of the trees at breast height (DBH). Mature forest plots were characterized by an average DBH >30 cm and a predominance of Abies religiosa, mixed with Pinus montezumae and P. hartwegii. Young forest plots were characterized by an average DBH ≤30 cm (c.f. Spies & Franklin, 1991; Lorimer, Dahir & Nordheim, 2001) and dominated by P. hartwegii. Though other attributes can be used to characterise forests (Weikel & Hayes, 1999; Huhta et al., 2004; Martin, Aitken & Wiebe, 2004), we used the common practice of defining forest type based on DBH and tree composition alone (Li & Martin, 1991; Miller, 2010).

Figure 1 Study location.

Aerial view of La Malinche (Google Earth), located in Tlaxcala State (insert). White polygons show our plots in mature (M1 M2, and M3) and young forest (Y1, Y2, and Y3). Map data: Google Earth, DigitalGlobe.

In 2009 we established one 24.25 ha study plot in each forest type (mature = M1; young = Y1; Fig. 1). In 2011 we increased our sample by setting up two more plots per forest type (mature = M2 and M3; Young = Y2 and Y3); these were somewhat smaller (16.5 ha each) to accommodate our recording schedule, and were composed of trees of different mean size than those in the 2009 plots. The 2011 plots in mature forest contained fewer very thick trees (mean DHB = 39.53 ± 11.20 cm) than the 2009 mature-forest plot (mean DBH = 66.03 ± 31.32), whereas the young forest plots, although still dominated by immature trees, contained more mature trees in 2011 (mean DBH = 31.20 ± 9.52 cm) than in 2009 (26.97 ± 8.14 cm; see Supplemental Information 1). Mean distance between plots was 4.3 km. Plots were located in the southern slope of La Malinche, at altitudes of 2,856–3,262 m.a.s.l. (Fig. 1). Within each plot we established sampling points 150 m apart (Fig. 2); plots established in 2009 had 12 points, and those from 2011 had eight points. At each point we recorded the altitude, measured height and DBH of every tree within a 25 m radius, and calculated tree density and fir/pine ratio. A discriminant function analysis based on DBH, tree height and altitude confirmed plot membership to either young or mature forest plots (F(6,69) = 13.4, Wilk’s λ = 0.4, P < 0.0001). The discriminant function correctly classified 93% of all sampling sites, and revealed significant differences between the forests that we deemed mature and those we classified as young (t = − 9.4, df = 54, P < 0.0001).

Figure 2 Study design.

Schematic representation of our plots showing the distribution of the 25-m radius count points within them. (A) Point counts in plots established in 2009 were carried out in three rows of four points each. In 2009, mature forest plots were supplemented with nest boxes at four points (dashed box), while young forests were supplemented at eight points (bold box). (B) In smaller plots established in 2011, point counts were conducted in two rows of four points each, and nest boxes were added at four of the eight points (bold box) in both forest types. Within each point, two nest box types were installed: (= standard boxes; = tree creeper boxes for Certhia).

Availability of cavities

In 2009, to determine the availability of natural cavities in both types of forest, we counted all cavities found in mature, dead or decaying trees at each point (tree categories as per Martin, Norris & Drever, 2006). We scanned 2,243 trees in the two study plots using 10 × 42 binoculars. This procedure, which is more expedient than following birds carrying nest materials or climbing trees to survey cavities (Cockle, Martin & Wiebe, 2008; Stojanovic et al., 2012), is conservative because it likely underestimates the number of cavities found in large trees and dense foliage, and small cavities are more difficult to detect in mature than in young forest (Koch, 2008). We considered a cavity to contain an active nest if we observed an adult entering and remaining inside the cavity for ten minutes or poking its head out on two or more occasions on different days. We also recorded the species of all cavity-bearing trees and their DBH. In 2010 we did not count cavities because we focused our efforts on installing and monitoring nest boxes. In 2011 we surveyed the cavities in the original (2009) plots, and measured, whenever it was accessible (at a height <2 m), the entrance height and diameter, and the width and the depth of the cavity. For new cavities, we also recorded the tree species and DBH.

Abundance of secondary cavity nesting birds

Twelve count points were homogeneously established within each of the 2009 plots (M1 and Y1) and eight points in each of the 2011 plots (M2, M3, Y2, Y3) at the intersections of a 200 × 200 m grid (Fig. 2). The points were located using a Garmin™ GPS (Bibby, Burgess & Hill, 1992; Ralph et al., 1996). In 2011 a row with four points was removed from the 2009 plots (M1 and Y1) to homogenize the sampling scheme (eight points per plot) that year. A minimum distance of 150 m between points was chosen to minimise the risk of counting individual birds more than once, based on the foraging behaviour of brown creepers during the breeding season, which takes place within ca. 100 m around the nest (Franzreb, 1985).

Each year (2009–2011) we conducted fortnightly bird counts from January to October. These were carried out from 8:00 to 13:00 h by following a set of pre-established routes that balanced both the time of day and the sequence in which different plots were visited. Point counts lasted ten minutes (Miller, 2010), and 15 min were allotted to move between points. During counts, every secondary cavity nester seen or heard within a 25 m radius was recorded (point-count survey methods followed (Bibby, Burgess & Hill, 1992; Ralph et al., 1996), see also (Manuwal & Huff, 1987; Martin, Norris & Drever, 2006) using a standardised surveying method that consisted of systematically scanning each tree within the observation radius from top to bottom using binoculars, starting at a haphazardly chosen direction and moving clockwise until completing a circle. This method maximised the probability of spotting secondary cavity nesters that normally forage on trunks (Franzreb, 1985). All counts were made by the same observer (CCL) and since each individual seen and/or heard was counted only once, each bird was deemed to be an independent observation at each point.

Nest box installation

In November 2009 and in January 2011 we installed nest boxes in all of our experimental study plots and monitored both the abundance of secondary cavity nesters and the breeding activity of those using nest boxes. To accommodate the preferences of all the secondary cavity nesting species present at La Malinche, we used two nest box designs; standard (intended to attract Sitta spp., Troglodytes aedon and Poecile sclateri), and tree creeper nest box (based on a design at www.birdfood.co.uk), to attract Certhia americana. All boxes were constructed from 1.5 cm thick pine plywood. Standard nest boxes had a forward-slanted roof, and measured internally 14.5 × 12 × 25 to 30 cm (width, depth, height), with a 3 cm diameter entrance hole placed at the middle of the frontal pane. Internally, the cuneiform boxes intended for tree creepers measured 13 × 12.5 × 35 cm (width, depth, height) with a 3 cm triangular entrance at the top of the (triangular) left side.

In 2009 we placed 80 nest boxes in eight of the 12 points in the young forest plot (Y1), and 40 boxes in four of the 12 points in mature forest plot, five boxes of each type were placed at each point (M1; Fig. 2A). We added twice as many boxes to young than to mature forest points so that cavity availability would be the same in both forest types (see results). Nest boxes, which were placed 10 m apart (Brush, 1983; Sánchez, Cuervo & Moreno, 2007), were fastened with a strap to a tree branch at the same height as natural cavities (between 5 and 10 m). Four of the 12 points in the young forest and eight of the 12 points in the mature forest did not receive nest boxes and acted as control points.

To allow comparison with other studies where an initial count of the birds was not carried out, we also placed nest boxes in the M2-M3 and Y2–Y3 plots in January 2011. Nest boxes were distributed in four of the eight points of each plot (Fig. 2B). Three boxes of each type were placed at each count point, resulting in 24 nest boxes per plot per vegetation type. Again, the remaining count points in each of the plots served as controls for the addition of nest boxes. In 2010 and 2011, between mid-February and early-September, all boxes were checked fortnightly, in the weeks when counts were not conducted, and weekly once evidence of nesting activity was detected (mid-March and early-August). In the latter case we recorded the species, number of eggs laid and number of fledglings produced of the birds using the nest box.

Permission to conduct this study was granted by the Mexican Ministry for the Environment (Secretaría de Manejo y Aprovechamiento de los Recursos Naturales; SEMARNAT, permit #SGPA/DGVS/04677/10).

Statistical analyses

Availability of cavities and abundance of secondary cavity nesting birds

Each point was deemed an experimental unit for the analyses (Ralph et al., 1996). We applied a χ2 homogeneity test to evaluate whether forest types differed in the number of trees with natural cavities, and a goodness-of-fit χ2 test to evaluate whether the total number of natural cavities differed between forest types.

Addition of nest boxes. We performed preliminary comparisons of total numbers of secondary cavity nesters recorded on each bird count at each point in both forest types (M1, Y1) in 2009 using a t-test (after verifying normality and equality of variances). Then, the effect of the addition of nest boxes was formally evaluated using a more complete approach that included the plots established in 2010–2011. We constructed a generalized linear mixed-effects model (GLMM) to detect whether the number of secondary cavity nesters was influenced by the addition of nest boxes in each forest type. Our model included forest type (young or mature) and treatment (with or without nest boxes), and their interaction as fixed effects. Random effects in the model were (1) point identity and (2) number of visits that we performed to each point (as this varied with breeding activity), both nested within year.

Due to the differences in the experimental design between 2009 and 2011, we implemented an additional test comparing the number of secondary cavity nesters observed in nest boxes versus control points in the plots added in 2011 (M2, M3, Y2, Y3), again using a GLMM. This model included forest type (young or mature) and treatment (with or without nest boxes) and their interaction as fixed effects. The number of visits to each point, nested within point identity, was included as a random effect. Our data showed signs of both excess of zeros and overdispersion with respect to a Poisson distribution; therefore, we constructed zero-inflated negative binomial GLMMs using the library glmmADMB for R (Bates, Maechler & Bolker, 2011; Skaug et al., 2011). All statistical analyses were carried out with R software, v. 3.1.0.

Results

Availability of cavities and abundance of secondary cavity nesting birds

Abundance of natural cavities

Young and mature forests had a similar number of trees with cavities (Table 1: χ2 = 2.2, df = 1, P = 0.14). However, as mature trees with cavities often had more than one, there were significantly more total cavities in mature than in young forests (goodness-of-fit tests contrasting with a distribution adjusted to the numbers of trees in both types of forest; χ2 = 9.5, df = 1, P = 0.002; Table 1). In 2011, cavity height was significantly higher in mature than in young forest (t = 11.0, df = 45, P < 0.0001), but had similar entrance diameter (t = 0, df = 39, P = 1), and depth (t = 1.5, df = 39, P = 0.16). Six cavities in mature forest were inaccessible and thus were not measured. These comparisons relate to attributes of natural cavities in 2011; most of the cavities measured that year were the same as those counted—but not measured—in 2009.

Table 1 Availability of natural cavities.

Numbers and characteristics of natural cavities found in the different types of forest each year (mean ± sd (min–max)).

Forest	2009	2011	
	Mature	Young	Mature	Young	
Number of trees with cavities	22	10	21	6	
Total number of cavities	50	17	37	10	
DBH of trees with cavities	60.90 ± 38.65	60.35 ± 16.95	97 ± 14.6	65.05 ± 24.95	
10.82–177.39	23.25–90.76	42.3–139.5	14.6–85.0	
Cavity height (m)	–	–	6.4 ± 1.5	1.14 ± 0.23	
1.83–10.2	1.11–1.17	
Entrance diameter (cm)	–	–	5 ± 0.33	5 ± 2.05	
4.3–5.6	3.5–9	
Vertical depth (cm)	–	–	9.5 ± 1.2	10.13 ± 0.67	
5.6–11	9.3–11.3	
Condition of tree with cavities					
Mature	1	6	4	3	
Snag	2	1	1	0	
Dead	19	3	16	3	
Species of tree with cavities					
Pine	21	19	21	6	
Oak	1	0	0	0	

In 2009, 13 of the 50 (natural) cavities found in mature forest were occupied (four by S. pygmea, five by T. aedon, two by P. sclateri, and two by C. americana). One of the 17 cavities located in young forest were occupied (by T. aedon). However, young and mature forest had similar proportion of cavities used (Fisher’s exact test: P = 0.095). Of the cavities originally found in 2009, thirteen in mature forest and seven in young forest were lost by 2011, mostly because the trees containing them fell. Of the 37 cavities identified in 2009 in the mature forest plots, five were again occupied by nesting birds in 2011 (two by S. pygmea, one by T. aedon, one by C. americana, and one by Colaptes auratus), and two new cavities were identified as nest sites (one occupied by S. mexicana and one by Melanerpes formicivorus). In the young forest, the cavity previously used in 2009 was again occupied in 2011 by nesting T. aedon (Fisher’s exact test: P = 1).

Bird counts

Although sightings were not abundant in either forest, we recorded a larger number of secondary cavity nesters in mature (x = 11.0 ± 7.0 [sd] birds per site across 20 visits) than in young forest (x = 5.3 ± 4.8; t = − 2.3, df = 22, P = 0.03) in 2009, consistent with our finding that there were more nesting cavities available in mature than in young forest.

Addition of nest boxes

The number of secondary cavity nester birds sighted increased following installation of nest boxes in both forest types (P = 0.05), and they appeared to be more abundant in the second year after the boxes were added (P < 0.01). Forest type had no effect on the number of secondary cavity nesters sighted after adding nest boxes (Table 2A; Fig. 3), and there was no interaction between forest type and treatment (addition of nest boxes; GLMM: Δ deviance2,10 = 0.168, P = 0.91).

Table 2 Results of generalized linear mixed-effects models (GLMMs).

The addition of nest boxes had a positive effect on the number of secondary cavity-nesting birds in both 2010 and 2011 (A) but forest type did not (B). We used a zero-inflated negative binomial error distribution.

Parameter	Estimate	Se	Z	P	
(A) Number of SCNs 2010–2011	
Intercept	−0.50	0.19	−2.65	<0.01	
Mature	0.21	0.16	1.34	0.18	
Boxes 2010	0.37	0.19	1.95	0.05	
Boxes 2011	0.99	0.19	5.20	<0.01	
(B) Number of SCNs in plots 2011	
Intercept	−0.47	0.23	−2.06	0.04	
Mature	0.15	0.17	0.88	0.38	
Boxes	0.95	0.17	5.58	<0.01	

Figure 3 Effect of nest box installation on number of birds.

Mean + SE number of secondary cavity-nesting birds (SCNs) in the study plots with or without nest boxes. SCNs were sighted more often at the points where nest boxes were added in both types of forest (P = 0.05), and they appeared to be more abundant in the second year after the boxes were added (P < 0.01).

In plots where nest boxes were added in 2011, there was a significant increase (P < 0.01) in sightings of secondary cavity nesters at points with nest boxes than at control points (Fig. 4). Again, neither forest type, nor the interaction of forest type with treatment (presence/absence of nest boxes) had an effect on the number of secondary cavity nesters seen (GLMM: Δ deviance1,8 = 1.842, P = 0.17; Table 2B).

Figure 4 Plots established in 2011 for each forest type.

In 2011, Mean + SE number of SCNs per count point in plots with or without nest boxes. SCNs were more often sighted in the points where nest boxes were added in both types of forest (P < 0.01).

Nest box usage

None of the 40 nest boxes added to the mature forest plots in 2009 was occupied in 2010, and only one was used by a pair of T. aedon in 2011. In contrast, 11 (14%) of the 80 boxes added to young forest in 2009 were used (one by C. americana, two by P. sclateri, two by T. aedon, and six by S. mexicana). The use of our nest boxes by S. Mexicana was unexpected, since the entrance to our boxes was deliberately made small to discourage species of this size. A similar number of boxes (12) was occupied in 2011 in young forest, again mostly by S. mexicana (n = 7), but also by P. sclateri (n = 2), and by S. pygmaea, S. carolinensis and T. aedon (one box each). After fledging one brood, the boxes used by P. sclateri and S. carolinensis were occupied by S. mexicana, and the box used by S. mexicana was subsequently occupied by T. aedon.

In plots incorporated in 2011, the picture was somewhat different; in the first year following nest box installation six out of the 48 boxes established in mature forest were occupied by T. aedon. Similarly, in young forest plots incorporated in 2011, ten out of the 48 boxes installed were used during the first year after installation (five by S. mexicana and five by T. aedon). In contrast to others studies (i.e., Jäntti et al., 2007), tree creepers did not use the nest boxes designed to accommodate their particular nesting habits. We only had one pair of Certhia americana occupying our boxes, and it nested in a standard nest box. Each nest box occupied represents one breeding pair, and all boxes had fledgling success >0, as we recorded no events of nest predation or usurpation, and no nests were abandoned that had settled in our boxes.

Discussion

There is continued interest in evaluating the availability and suitability of nesting sites in different environments because these often limit the local diversity of species and the number of breeding pairs (Newton, 1994; Martin, Aitken & Wiebe, 2004; Wesolowski, 2007; Cockle, Martin & Drever, 2010). Before installing artificial nest boxes, we found that secondary cavity nester sightings were more frequent in mature than in young forest (P = 0.03), perhaps related to the availability of natural cavities. We also found a higher percentage of natural cavities occupied by secondary cavity nesters in mature than in young forest, although the difference was not significant. This figure (26%) is much lower than those from northern temperate forests (61–93% Van Balen et al., 1982; 67% Ingold & Ingold, 1984; 57% Peterson & Gauthier, 1985), but higher than in neotropical habitats, where decay promotes the production of cavities- (c.f. 5% in Argentinean Atlantic forest (Cockle, Martin & Wiebe, 2008); 2% in Peruvian Amazon forest (Brightsmith, 2005)). Although located well within the tropics, the mature forest at La Malinche is more similar in both climate and biological community to northern temperate forests than tropical forests, thus the low proportion of occupied cavities may indicate a low population density of secondary cavity nesters, rather than a superabundance of cavities as seen in warmer and moister forests.

Breeding populations of cavity nesting birds may be limited by other factors including territoriality and interspecific competition for nest sites (Newton, 1994; Newton, 1998; Dhondt, 2012) or sites vulnerable to predation (Nilsson, 1984). There is no information on territory size or territorial behaviour of any bird species at La Malinche, thus we cannot rule out the possibility that the small proportion of cavities used in the mature forest is due to territoriality. However, we think that this is unlikely because adjacent nest boxes in young forest were often occupied, both by the same and by different species. Although some nest boxes were used twice by the same or different species (n = 3), we did not observed any agonistic behaviour or other forms of inter- or intra-specific usurpation.

Cavity abundance alone may be insufficient to explain cavity use; as Lohmus & Remm (2005) have argued, cavity quality (e.g., height, size, depth, degree of tree decay, etc.), together with abundance, determine the real availability of appropriate nesting holes (see also Rendell & Robertson, 1989; Wiebe & Swift, 2001; Cockle, Martin & Wiebe, 2008; Cockle, Martin & Robledo, 2012). We do not have sufficient data to estimate the proportion of natural cavities that are suitable for nesting in the mature forest. However, over the two years when nest boxes were available, only one was occupied in the mature forest, thus scarcity of suitable cavities does not appear to be limiting cavity use by secondary cavity nesters in that forest. Conversely, the significant increase of sightings of secondary cavity nesters in the young forest following the addition of nest boxes indicates that in this habitat nest site availability is indeed a limiting factor for the populations of those birds.

The higher occupancy of nest boxes in young forest (31.95%) compared to the mature forest (8.3%) was to a degree driven by the readiness with which S. mexicana took to breeding in them. This is similar to the findings of Miller (2010) that bluebirds readily colonise forest gaps, forest edges, and other plant communities with few trees, as is true in the La Malinche young forest. However, another five species successfully nested in the boxes that we provided, thus the benefits of this practice extended to their populations. Given that substantial sections of La Malinche are occupied by young forest, it is likely that the population size of secondary cavity nesters is much smaller than this habitat could potentially maintain.

Our data provide additional support for the hypothesis that in managed/secondary forest, the populations of secondary cavity-nesting birds are limited by the scarcity of cavities (Brawn & Balda, 1988; Waters, Noon & Verner, 1990; Tomasevic & Estades, 2006). As pristine environments disappear and natural woodland gives way to managed forest, the structure of its biological communities will largely depend on the decisions we make to protect and increase biological diversity (Janzen, 1998). One way in which we can contribute to promoting community richness in managed forests is by supplementing key resources that are lacking or scarce in those environments, such as appropriate cavities for nesting. These, in addition to increasing the density of insectivorous cavity nesting birds, can promote the fitness of the plants in which they forage by substantially reducing the number of insects on the plants (Sanz, 2001). Forest management practices that promote the conservation of insectivorous birds are fundamental for the maintenance of forest productivity, through controlling of the populations of pest insects (Marquis & Whelan, 1994).

Forest management programs throughout Latin America currently do not, but should, include the addition of nest boxes to forest plantations. Because nest boxes increase bird numbers and species diversity, their addition should be encouraged to generate the benefits associated with a richer bird community.

Supplemental Information

File S1 Tree size by type of forest

DBH of trees in the study plots of mature and young forest sampled in 2009 and 2011.

Click here for additional data file.

File S2 Numbers of birds counted

Numbers of birds counted on each plot in each visit.

Click here for additional data file.

File S3 Measurements of all the trees included in the analyses

Height, species, and DBH of all trees entered in the study, organised per plot and year.

Click here for additional data file.

Supplemental Information 1 Location of experimental plots

List of geographical coordinates of the sampling points.

Click here for additional data file.

Several undergraduate students from the Tlaxcala (UAT), Puebla (BUAP) and National (UNAM) autonomous universities helped in different phases of the fieldwork. We thank A Salinas-Melgoza and A Ríos-Chelén for their feedback on drafts, and G Moreno-Rueda and an anonymous referee further helped improving the manuscript. L Kiere and O Sánchez-Macouzet advised on the statistical analyses. CCL was supported by a scholarship from CONACyT (45901) to conduct PhD studies in the Doctorado en Ciencias Biomédicas, Universidad Nacional Autónoma de México; the present paper is submitted in partial fulfilment of the program’s requirements.

Additional Information and Declarations

Competing Interests

Author Contributions

Animal Ethics

Field Study Permissions

Data Availability

The authors declare there are no competing interests

Cecilia Cuatianquiz Lima conceived and designed the experiments, performed the experiments, analyzed the data, wrote the paper, prepared figures and/or tables, reviewed drafts of the paper, provided equipment and logistics.

Constantino Macías Garcia conceived and designed the experiments, analyzed the data, contributed reagents/materials/analysis tools, wrote the paper, reviewed drafts of the paper, provided equipment and logistics.

The following information was supplied relating to ethical approvals (i.e., approving body and any reference numbers):

We conducted an experiment that consisted on adding nest boxes and counting birds either inhabiting different sectors of the forest or nesting in the boxes provided. We did not manipulate the birds.

The following information was supplied relating to field study approvals (i.e., approving body and any reference numbers):

Permit to conduct the field experiment was granted to CMG by the Mexican Ministry for the Environment (SMARNAT, permit #SGPA/DGVS/04677/10).

The following information was supplied regarding data availability:

All the raw data files were provided as Supplemental Information 1.

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
