# Peer review of "Pre- and post-experimental manipulation assessments confirm the increase in number of birds due to the addition of nest boxes"

_PeerJ, doi:10.7717/peerj.1806_

## Round 0.1 · original submission · Major Revisions

· Academic Editor

Major Revisions

It is recommended that you thoroughly revise the experimental design part, and follow the suggestions provided by the reviewers.

·

Basic reporting

No cmments, it is all right.

Experimental design

The study was apparently carried out according to the ethical standards, the design was excellent, and rigoruously performed. Methods and goals of the study are well defined.

Validity of the findings

Data are robust, the statistical analysis is excellent. Conclusions are appropriate.

My only point is in lines 215-223, where a statistical test is lacking.

Comments for the author

This is an excellent study, well designed, well analysed, and well written. I only have some minor comments, mainly indicating some errata.

L. 34-38: In general, the paper shows long lists of references, which are repeated several times. Probably, authors are interested in Casas-Crivillé & Valera (2005; J Arid Environm 60:227), as an example of the importance of a primary cavity nester as founder of sites for other species, secondary cavity nester.

L. 64: Nest-boxes are used also by insects and other invertebrates, mainly paper wasps.

Introduction: note that the use of nest-boxes not only has positive effects on birds, but also positive effects on plants, and the complete ecosystem by promoting the populations of insectivorous birds. See e.g. MARQUIS, R. J. & WHELAN, C. J. 1994. Insectivorous birds increase growth of White oak through compsumption of leaf-chewing insects. Ecology, 75: 2007-2014.

L. 89: erratum: Americana should be in minuscule.

L. 130: > or <?

L. 215-223: I would like to see a statistical test comparing the use of cavities between the two types of forests.

L. 236: please, put the statistic result, although it is in the figure legend.

L. 243: the mistake twice: Mexicana should be in minuscule.

L. 260: Are you sure? I don’t see a test supporting this contention, although data obviously support it.

L. 266: That’s right, the fauna reported is holarctic.

Table 1: review, it seems that a raw was moved.

This is an excellent study, whose results will be useful for bird conservation.

Dr. Gregorio Moreno-Rueda
Universidad de Granada
Spain

Reviewer 2 ·

Basic reporting

Lima and Garcia
In this paper the authors explore if availability of cavities affects the number of secondary cavity nesters (SCN) in two forest types in Mexico by providing artificial nest boxes and comparing bird communities before (1 year) and after (2 years).
Lines 43: as to the limititation caused by absence of natural cavities: see Chapter 5 in Dhondt 2012 (Interspecific competition in Birds) for a review, and chapter 8 for effects of manipulating nest boxes (line 65).
Line89: I believe that in the name Certhia americana, americana should be lower case.
Line 97: I am surprised that trees with a DBH of 30 cm would be considered mature.
108: sampling points 200 m apart. According to fig. 2 they are 150 m apart – explain or correct

112: you define young forest as having trees with dbh <30 cm; however the average is 29.7 with a SD of 12.6 cm. Clearly a large number of trees have a dbh >30 cm. confusing.

136: I am not sure why you remove 4 points from the 2009 plots. You could calculate the number of birds/ha
150: “the normally trunk dwelling cavity nesters” confusing. I assuming you mean the cavity nesters that normally forage on trunks.
Nest box installation: poorly explained – please make an effort to describe clearly.
160: different n umbers of boxes in different habitats could bias the results if species prefers a box over a natural cavity
162: having boxes 10 m apart is strange: clearly in territorial species the same species would not normally breed so close together.
155, 165: placing boxes in November could entice birds to settle and use boxes as roosting sites, while this would not happen when placed in January. The effect of adding boxes, therefore, could be different.
169: you do not provide any information on the size of the next boxes, nor the very important information as regards the size of the entrance opening. Also, I am not sure what conical boxes are. I know that Finnish scientists designed “boxes” for treecreepers (see for ex. Jantti ea 2001, 2007; Aho ea 1997) that were very successful.
172: I am not sure over what period you checked the boxes every other week.
174: did you determine laydate?
185: I am not sure what numbers you compared: mean per point?
210, table 2: you report significance of results, but not the results themselves. Thus it would be useful to say what cavity height actually was.
Fig 3: I assume the graph represents average number per point?
234: Mexicana : lower case
245: here Mexicana is lower case – be consistent
250: you write “occupied”. Does this means the birds were nesting, laid eggs? In wrens males build a part of the nest, that females, when they accept it, finish and lay eggs. Given the fact that you boxes were very close together at each point, if several boxes contained nest, but no eggs, this might simply be one male. This inflates the result.
290: bluebirds nest in more open habitat. The results suggest that young plots were quite different from mature ones, thereby attracting bluebirds. Was the species composition different between young and mature plots?

Experimental design

poorly describe - see above

Validity of the findings

could be better.
report the results in more detail

Comments for the author

Potentially interesting paper - describe results in more detail.

---

## Round 0.2 · accepted · Accept

· Academic Editor

Accept

No further comments, revision was accepted by the reviewers

·

Basic reporting

The authors satisfactorily have respond my concerns. I have no ulterior comments.

Experimental design

Ok.

Validity of the findings

Ok.